# Co-Existence of Atomic Pt and CoPt Nanoclusters on Co/SnO_x_ Mix-Oxide Demonstrates an Ultra-High-Performance Oxygen Reduction Reaction Activity

**DOI:** 10.3390/nano12162824

**Published:** 2022-08-17

**Authors:** Amisha Beniwal, Dinesh Bhalothia, Wei Yeh, Mingxing Cheng, Che Yan, Po-Chun Chen, Kuan-Wen Wang, Tsan-Yao Chen

**Affiliations:** 1Department of Engineering and System Science, National Tsing Hua University, Hsinchu 30013, Taiwan; 2Department of Materials and Mineral Resources Engineering, National Taipei University of Technology, Taipei 10608, Taiwan; 3Institute of Materials Science and Engineering, National Central University, Taoyuan City 32001, Taiwan; 4Hierarchical Green-Energy Materials (Hi-GEM) Research Centre, National Cheng Kung University, Tainan 70101, Taiwan

**Keywords:** oxygen reduction reaction, fuel cells, Pt-utilization, mass activity, potential synergism

## Abstract

An effective approach for increasing the Noble metal-utilization by decorating the atomic Pt clusters (1 wt.%) on the CoO_2_@SnPd_2_ nanoparticle (denoted as CSPP) for oxygen reduction reaction (ORR) is demonstrated in this study. For the optimum case when the impregnation temperature for Co-crystal growth is 50 °C (denoted as CSPP-50), the CoPt nanoalloys and Pt-clusters decoration with multiple metal-to-metal oxide interfaces are formed. Such a nanocatalyst (NC) outperforms the commercial Johnson Matthey-Pt/C (J.M.-Pt/C; 20 wt.% Pt) catalyst by 78-folds with an outstanding mass activity (MA) of 4330 mA mg_Pt_^−1^ at 0.85 V vs. RHE in an alkaline medium (0.1 M KOH). The results of physical structure inspections along with electrochemical analysis suggest that such a remarkable ORR performance is dominated by the potential synergism between the surface anchored Pt-clusters, CoPt-nanoalloys, and adjacent SnPd_2_ domain, where Pt-clusters offer ideal adsorption energy for O_2_ splitting and CoPt-nanoalloys along with SnPd_2_ domain boost the subsequent desorption of hydroxide ions (OH^−^).

## 1. Introduction

The intense exploitation of traditional fossil fuels and their associated environmental impacts are prompting researchers to explore renewable and sustainable energy devices with low carbon emissions. In this regard, fuel cells have emerged as potential contenders owing to their independence from fossil fuels and zero carbon emissions [1,2]. Even after possessing prodigious traits, the commercial viability of fuel cells is primarily restricted by the intrinsically sluggish kinetics of oxygen reduction reaction (ORR) at the cathode side, which occur at the highest energy barrier (~0.3 to 0.4 V) compared to hydrogen oxidation reaction (~0.1 V) at anode side and hydroxide ion (OH^−^) diffusion (~0.05 V) through the membrane [3]. The ORR determines the overall performance of fuel cells and demands highly efficient electrocatalysts to overcome the gigantic overpotential. By far, Platinum (Pt)-based catalysts are the most prevailing ORR catalysts [4,5]. However, limited availability and whopping cost are the major stumbling blocks associated with Pt making it unsuitable for commercial applications [6]. On top of that, fuel crossover effects, CO poisoning, time-dependent drift, poor selectivity of intermediate ORR species (OH*, OOH*, and O*), and material longevity at counterpart electrodes in critical redox conditions are major issues holding up the performance of Pt-based catalysts at the fuel cell cathode [7]. Despite various notable assessments, none other than Pt has been commercialized. By keeping such scenarios in view, increasing Pt utilization with minimal loading and maintaining a decent power output is pivotal for the broad commercialization of fuel cells. Encouraging progress over the timeline has been demonstrated in this direction [8,9]. More specifically, Pt–M (where M corresponds to the transition metals) nanostructures (alloys or Pt-shell@M-core) are at the forefront and have been extensively studied in the past several decades [10,11]. These nanostructures deliver practical catalytic performance with reduced Pt content. By taking advantage of the critical interplay between electronic effect (i.e., ligand) and geometric strain, the Pt–M nanostructures alter the binding energies of ORR intermediates on their surface; hence, the ORR performance of such materials is improved as compared to carbon-supported Pt nanoparticles [12,13]. However, the unevenly distributed surface-active sites and unavoidable composition leaching in harsh redox conditions are key issues detaining Pt-alloys for their wide-scale application [14]. Moreover, the geometric strain and electronic ligand effects are strongly dependent on the thickness of the Pt shell and progressively reduced with the increasing number of Pt layers [15]. In this event, Pt-monolayer over M-core seems to be the optimum design. However, Pt atoms are the sole identities to promote ORR on the catalyst surface, and so the selectivity of transitional reaction steps between surface sites in these core-shell structures is subdued. On top of that, the minimal metal loading constraints to form Pt-monolayer and lack of delicate fabrication techniques are ruling out the application of core-shell nanostructures for practical fuel cell applications. The interfaces of heterogeneous nanocatalysts (NC)s are considered to be active sites for catalytic reactions [16,17]. In this context, the formation of multiple interfaces in sub-nanometer domains can be a possible solution for utilizing the lattice strain and ligand effect simultaneously in heterogeneous NCs with a size range of 3–5 nm. Both of these effects together shift the d-band center relative to the Fermi level and thus manipulate the chemisorption strength of ORR intermediates. Although the interfaces can promote the electrocatalytic activity of heterogeneous NCs, however, the minimal metal loading constraint is still there. Consequently, designing high-performance ORR NCs with reduced loading and increased Pt-utilization is imperative. To this end, anchoring atomic Pt-clusters at the surface and interface of Pd-based multi-metallic heterogenous NC with metal-to-metal-oxide interfaces is an efficient assessment to minimize noble metal usage [18]. Palladium (Pd) possesses nearly similar physicochemical properties as that of Pt [19]. In addition, Pd exhibits better CO tolerance and is more abundant relative to Pt. For instance, our previous works demonstrated that surface anchored Pt or Au-clusters assist in the synchronous operation of the intermediate steps, resulting in the quantum leap in the ORR performance of NCs [20,21,22,23]. In addition, single atoms catalysts and transition metal-based catalysts have also been frequently reported [24,25,26].

By keeping the aforementioned scenarios in view, herein, we prepared quaternary NCs consisting of Pt-clusters decorated CoO_2_@SnPd_2_ hierarchical structure (namely CSPP) via a temperature-controlled wet chemical reduction method. For the optimum condition, when the impregnation temperature for Co-crystal growth is 50 °C, the CoPt nanoalloys and Pt-clusters decorated CoO_2_@SnPd_2_ hierarchical structure with multiple metal-to-metal oxide interfaces is formed (hereafter denoted as CSPP-50), which surpasses the commercial Johnson Matthey-Pt/C (J.M.-Pt/C; 20 wt.% Pt) catalyst by 78-folds with an outstanding mass activity (MA) of 4330 mA mg_Pt_^−1^ at 0.85 V vs. RHE in alkaline ORR (0.1 M KOH). On the other hand, Pt-clusters emerged SnO_2_ layer encapsulated CoO_2_@SnPd_2_ and Sn-oxide layer enclosed CoO_2_@CoSnPdPt alloy structures are formed at 25 °C (CSPP-RT) and 75 °C (CSPP-75) impregnation temperatures, respectively. The MA values for CSPP-RT and CSPP-75 NCs are 2623 mA mg_Pt_^−1^ and 779 mA mg_Pt_^−1^, respectively, exhibiting much suppressed ORR performance as compared to CSPP-50. The cross-referencing results of physical characterizations and electrochemical analysis suggest that the surface anchored Pt-clusters and CoPt-nanoalloys play a key role in the high ORR performance of CSPP-50 NC, where Pt-clusters offer ideal adsorption energy for O_2_ splitting and CoPt-nanoalloys along with Sn_x_Pd_y_ domain boost the subsequent desorption of hydroxide ions (OH^−^).

## 2. Experimental Procedures

### 2.1. Preparation of CSPP Nanocatalysts

Pt-clusters decorated CoO_x_@Sn_x_Pd_y_ quaternary NCs (denoted as CSPP) were synthesized via sequential wet chemical reduction method with temperature control during CoO_x_-core crystal growth. Prior to nanoparticle growth, the surface functionalization of catalyst carriers (i.e., carbon nanotubes (CNT)s, Cnano Technology Ltd., Zhenjiang, China) was carried out as per reported in our previously published studies for better attachment and uniform dispersion [27]. After surface functionalization, the CNTs were immersed in Deionized (DI) water to prepare a solution (solution A) with a content of 5% (*w*/*w*). Furthermore, 1200 mg of solution A (i.e., the actual amount of CNTs is 60 mg) was mixed with 3.06 g of aqueous solution of 0.1 M cobalt (III) chloride trihydrate (CoCl_3_·3H_2_O, 99%, Sigma-Aldrich Co., St. Louis, MO, USA) and stirred at 400 rpm at 25 °C, 50 °C and 75 °C for 4 h (Step-1st). The mixture (i.e., Co^3+^ adsorbed CNT, CNT-Co^3+ ads^) contains 0.306 mmoles (18 mg) of Co metal ions (Co^3+^) with a weight ratio of 30 wt.% as that of CNT. After stirring, 5 mL of D.I. water solution consisting of 0.11 g of sodium borohydride (NaBH_4_; 99%, Sigma-Aldrich Co.) was instantly added to the solution prepared in the 1st step (i.e., Co^3+^ adsorbed CNT, CNT-Co^3+ ads^) and stirred at 400 rpm for 10 s (Step-2nd). In this step, metastable Co metal NPs were formed on the CNT surface, which later partially oxidized and turned into a mixed metallic and oxide phase (i.e., Co/CoO_x_) (solution B). In the 3rd step, a solution containing 7.12 g of Pd^2+^ ions adsorbed Sn(OH)_x_) with the ratio of Pd/Sn = 1.0 was added into solution-B, where Sn^2+^ and Pd^2+^ ions are reduced by the excessive amount of NaBH_4_ added in the 2nd step. In this step, the Sn and Pd atoms formed Sn_x_Pd_y_ alloy and hierarchically deposited on the CoO_x_ core-crystal (i.e., CoO_x_@Sn_x_Pd_y_; denoted as CSP) (solution-C). For preparing the Pd precursor, the Pd metal powder (PdCl_2_, 99%, Sigma-Aldrich Co.) was dissolved in 1.0 M HCl (aq) by stirring at 400 rpm at 25 °C, whereas the 0.1 M Sn precursor solution was prepared via mixing the Tin (II) chloride (SnCl_2_, 99%, Sigma-Aldrich Co.) in D.I. water. The atomic ratio of Co-Sn-Pd in the CSP NC is 1:1:1. Following the preparation of CSP NCs, the decoration of atomic Pt-clusters was achieved by the reduction of Pt^4+^ ions on the surface and interface of CSP NCs. Prior to decoration, as-prepared CSP NCs were subjected to an ultrasonication treatment for creating the defect sites at the surface and interfaces to accommodate the Pt clusters. Subsequently, 0.007 g of 0.1 M Pt precursor solution was added to solution “C” (i.e., CoO_x_@Sn_x_Pd_y_; CSP) and reduced by using NaBH_4_. The end products are CSPP NCs with different configurations but the same composition. As-prepared CSPP NCs were washed several times with acetone, centrifuged, and then dried at 100 °C overnight. Hereafter, the CSPP NCs are denoted as CSPP-RT, CSPP-50, and CSPP-75, respectively, for samples impregnated at 25 °C, 50 °C, and 75 °C temperatures.

### 2.2. Physical Inspections of CSPP Nanocatalysts

The actual compositions of the experimental samples were determined by an inductively coupled plasma-optical emission spectrometer (ICP-OES, Agilent 725, Santa Clara, CA, USA). The cross-referencing results of electron microscopy and X-ray spectroscopy were utilized to unveil the physical properties of CSPP NCs. First, the high-resolution transmission electron microscope (HRTEM) images were collected at the electron microscopy center of National Tsing Hua University, Taiwan. The HRTEM samples were prepared on the 200 mesh copper grids. The contaminated species on the specimen’s surface were cleaned by plasma treatment before loading into the HRTEM chamber. Furthermore, X-ray diffraction (XRD) was carried out at National Synchrotron Radiation Research Center (NSRRC), Taiwan (beamline of BL-01C2), with an incident X-ray of wavelength 0.688 Å and energy 18 KeV. Moreover, the X-ray absorption spectroscopy (XAS) of experimental NCs was executed at beamlines BL-17C and 01C1 of NSRRC, Taiwan. For acquiring the acceptable quality spectra, each measurement was repeated twice and averaged for successive comparisons. The energy calibration and normalization of as obtained XAS spectra were performed by using ATHENA software, whereas ARTEMIS software was used for the extended X-ray absorption fine spectra (EXAFS) analysis. For EXAFS analysis, the normalized spectra were transformed from energy to k-space and further weighted by *k*^3^ to distinguish the contributions of backscattering interferences from different coordination shells. Normally, the backscattered amplitude and phase shift functions for specific atom pairs were theoretically estimated utilizing the FEFF8.0 code. The ICP-AES (Jarrell-Ash, ICAP 9000) was used for the exact atomic compositions of the CSPP NCs.

### 2.3. Electrochemical Analysis of CSPP Nanocatalysts

The electrochemical analysis of CSPP NCs has been carried out as per the previously reported method [28]. In detail, a potentiostat (CH Instruments Model 600B, CHI 600B) equipped with a three-electrode system was used for the electrochemical measurements. The cyclic voltammetry (CV) and linear sweep voltammetry (LSV) data were obtained at the voltage scan rate of 0.02 V s^−1^ and 0.001 V s^−1^, the potential range of 0.1 V to 1.3 V (V vs. RHE) and 0.4 V to 1.1 V (V vs. RHE), respectively in an aqueous alkaline electrolyte solution of 0.1 M KOH (pH 13). The N_2_ and O_2_ atmospheres were used for CV and LSV, respectively. All the measurements were carried out at room temperature. The detailed procedure catalysts slurry preparation and the ORR mass activity calculation have been given in Appendix A.

### 2.4. Preparation of the Electrode and the Calculation for ORR Mass Activity

The catalyst slurry for ORR test was made by dispersing 5.0 mg catalyst powder in 1.0 mL of isopropanol containing 50 μL of Nafion-117 (99%, Sigma-Aldrich Co.). This mixture was subjected to ultra-sonication for 30 min prior to the ORR test. For conducting ORR test, 10.0 μL of catalyst ink was drop cast and air-dried on a glassy carbon rotating disk electrode (RDE) (0.196 cm^2^ area) as working electrode. Hg/HgCl_2_ (the voltage was calibrated by 0.242 V, in alignment with that of RHE) electrode saturated in KCL aqueous solution and a platinum wire were used as reference electrode and counter electrode, respectively.

The kinetic current density (*J_K_*) and the number of electrons transferred in ORR were calculated based on the equations below:$$\frac{1}{J}=\frac{1}{J_{K}}+\frac{1}{J_{L}}=\frac{1}{J_{K}}+\frac{1}{B\omega^{0.5}}\;B=0.62nFC_{O_{2}}D_{O_{2}}{}^{\frac{2}{3}}\nu^{-\frac{1}{6}}$$
where *J*, *J_K_* and *J_L_* are the experimentally measured, mass transport free kinetic, and diffusion-limited current densities, respectively. *ω* is the angular velocity, *n* is transferred electron number, F is the Faraday constant, *C_O_*_2_ is the bulk concentration of O_2_, *D_O_*_2_ is the diffusion coefficient, *v* is the kinematic viscosity of the electrolyte. For each NC, the mass activity was obtained when *J_K_* was normalized to the Pt loading.

The *mass activity* (MA) is calculated by the following equation
$$mass\;activity\;\left({\mathrm{mA}\;\mathrm{mg}}^{-1}\right)=J_{K}\times \frac{area}{mass\;of\;catalyst}$$
where *J_K_* is the kinetic current density (mA/cm^2^) and the *area* is the geometric area of the working electrode (0.196 cm^2^). The mass activity of the catalyst is estimated via the calculation of *J_K_* and normalization to the catalyst loading on the glassy carbon rotating disk electrode.

## 3. Results and Discussion

### Surface Morphology and Crystal Structure of CSPP Nanocatalysts

The actual composition of the as-prepared CSPP NC is determined by ICP-OES, and the corresponding results are listed in Appendix A. The structure and morphology of the as-prepared CSPP NCs were investigated via HRTEM. Figure 1 shows the HRTEM images of CSPP NCs prepared at different impregnation temperatures, whereas corresponding Inverse Fourier Transformed (IFT) images (shown in upper right corners) and line histogram (shown in insets) of selected fringes are used to calculate the d-spacing. Figure 1 confirms that variable impregnation temperature during Co-core crystal growth had a significant effect on the crystal structure and overall morphology of CSPP NCs. Such a variation in the surface, as well as the sub-surface configuration of CSPP NCs, can be attributed to the different proportions of atomic interdiffusion/alloying (i.e., heteroatomic intermixing). First, the HRTEM image of CSP NC (i.e., without Pt decoration) is discussed to clarify the structural and morphological differences after the surface and interface decoration of Pt-clusters. As shown in Appendix A, a fuzzy thin layer (denoted by red triangles) encapsulated by disordered structures (denoted by white circles) is observed in CSP NC, which can be assigned to the formation of SnPd_2_ intermetallic alloys due to the intercalation of Pd-atoms in the thin layer of SnO_2_ over CoO_2_ crystal underneath (i.e., CoO_2_@SnPd_2_@SnO_2_). The formation of such a complex structure is consistently confirmed by the X-ray spectroscopic techniques in the following sections. Herein, the surface capped SnO_2_ layer blocks the Pd-active sites. Hence, the ORR performance of CSP NC is expected to suppress. Furthermore, varying the impregnation temperature during Co-core crystal growth and anchoring of atomic Pt clusters at the surface and interface substantially changed surface as well sub-surface configurations of CSP NC.

Figure 1a depicts the HRTEM image of CSPP-RT NC, where it is evident that the CSPP-RT NCs (denoted by yellow circles) are grown on the CNT support (denoted by the white line) with nearly spherical morphology. For CSPP-RT NC, the Co-atoms are impregnated with CNTs at room temperature, where the majority of Co-atoms are expected to agglomerate in the form of small particles due to relatively slow atomic migration. Subsequently, when a mixture of Pd^2+^ and Sn^2+^ ions (i.e., Pd^2+^ adsorbed Sn(OH)_x_) are added to the reaction system, the SnPd_2_@SnO_2_ layers over the CoO_2_ core-crystal underneath are formed. Given that the Pt loading in CSPP NCs is 1.0 wt.%, the galvanic replacement reaction is severely limited. In this event, the Pt-atoms are expected to accommodate only in the highest defected region of the topmost SnO_2_ layer with a strong lattice mismatch. Such a scenario can be confirmed by the highly disordered surface morphology and dislocations in IFT patterns (denoted by red circles) of CSPP-RT NC. The random interlayer spacing (i.e., 0.218 nm and 0.261 nm) complementary proves the strong lattice mismatch between Pt and SnO_2_ domains. Figure 1b,c, respectively, shows the representative HRTEM images of CSPP-50 and CSPP-75 NCs. Accordingly, the clearly observed lattice fringes suggest that the crystallinity of CSPP-50 NC is greatly improved in the surface as well as sub-surface regions and can be attributed to the removal of a thin layer of SnO_2_. (Figure 1b) For CSPP-50, the increased crystalline properties in both surface and sub-surface regions can be attributed to the exposure of Pd and/or Pt atoms. At the increased impregnation temperature of Co-core crystal growth, the increased kinetics of galvanic replacement reaction between Pd^2+^ <-> Co followed by Pt^4+^ <-> Co/Pd ions is obvious. In this event, the Co atoms are exposed to the surface and form the CoPt nanoalloys along with Pt-clusters on the surface of CSPP-50 NC, which can be confirmed by the slightly disordered regions (denoted by white circles) on the surface of CSPP-50 NC and consistently confirmed by the XRD and XAS analysis in following sections. Meanwhile, the slight lattice dislocations (denoted by the red square) in the sub-surface region indicate the formation of SnPd_2_ alloy. These characteristics are further confirmed by the FFT analysis, where the co-existence of two asymmetric FFT spots (a-a* and b-b*) with significant intensity and length difference can be attributed to the presence of Pt-clusters and CoPt nanoalloys on the surface with strong lattice strain. Moreover, the unevenly distributed FFT spots in outer space (denoted by red circles) can be assigned to the formation of SnPd_2_ alloys in the subsurface region. All these observations integrally suggest the formation of CoPt nanoalloys and Pt clusters decorated SnPd_2_ layer over CoO_2_ crystal underneath. Furthermore, Figure 1c shows the HRTEM image of CSPP-75 NC, where multifaceted, highly crystalline regions (denoted by yellow triangles) were observed in the inner lattices along with a thick layer of SnO_2_ on the surface (denoted by red triangles). The formation of such a structure can be attributed to the presence of multi-metallic alloys (i.e., CoSnPdPt) in the subsurface region due to the increased atomic migration and higher kinetics of galvanic replacement reaction at higher impregnation temperature. In this case, SnO_2_ enclosed CoSnPdPt alloy formation is expectable.

The XRD analysis was employed to delve more deeply into the crystal structure of CSPP NCs. First, the crystal structure of CSP NC is elucidated by comparing the XRD spectra of CSP NC with control samples (Co-CNT, Sn-CNT, and Pd-CNT), while the average coherent lengths of samples are summarized in Appendix A As shown in Appendix A, Co-CNT exhibits several intense peaks (denoted by black triangles) corresponding to the different facets of the CoO_2_ phase. The Co@Pd NC exhibits the diffraction signals for Pd (peaks C and D) as well as for CoO_2_ (peaks X and Y), suggesting the formation of an incomplete Pd shell over CoO_2_ underneath. Moreover, the presence of substantially broad peak E confirms the formation of SnPd_2_ alloy in Sn@Pd NC. An even closer inspection reveals that the peak E is shifted to the left side as compared to peak D of Pd-CNT, suggesting the lattice relaxation in Pd domains and is a clear indicator for alloy formation. On the other hand, the peak F* is shifted to a higher angle as compared to peak F of Sn-CNT and can be attributed to the lattice compression in Sn-domains due to the formation of Sn-oxide. On top of that, the CSP NC exhibits the characteristics corresponding to the SnPd_2_ alloy (hump A), SnO_2_ (intense peaks denoted by pink squares), and CoO_2_ (relatively suppressed peaks denoted by black triangles). All these observations integrally indicate the formation of CoO_2_-supported SnPd_2_ alloys with a thin layer of SnO_2_ on the surface. Furthermore, as shown in Figure 2, the CSPP-RT NC exhibits a nearly similar XRD pattern as that of CSP NC, where the absence of Pt diffraction signals confirms that decorated Pt species emerged in the top most SnO_2_ layer. Moreover, CSPP-50 NC exhibits a hump (denoted by a half circle) from 17° to 18° and can be attributed to the decorated Pt-clusters. Moreover, the increasing intensity of diffraction peaks (denoted by triangles) indicates the improving crystallinity of CoO_2_ and exposure to the surface. In addition, the peaks M, N, and O are observed for CSPP-50 NC, which respectively correspond to the CoPt (101), CoPt (110), and CoPt (002), confirming the formation of CoPt nanoalloys in the CSPP-50 NC. These scenarios are further confirmed by the offset of CoO_2_ peaks to lower angles, which suggest the lattice relaxation due to CoPt alloy formation. These observations confirm the formation of CoPt nanoalloys and Pt clusters in the CSPP-50 NC and are in good agreement with former HRTEM analysis and consistently proved by XAS analysis in the subsequent section. Meanwhile, CSPP-75 NC mostly exhibits the diffraction signals corresponding to the SnO_2_ and CoO_2_, suggesting the formation of CoO_2_ and/or SnO_2_ layer on the surface.

To obtain more insights into the variations in the local atomic arrangements and electronic structure of CSPP NCs with varying impregnation temperature, the XAS analysis at Pt L_3_-edge, Sn K edge, and Co K-edges were conducted. Figure 3a shows the normalized Pt L_3_-edge X-ray absorption near edge (XANES) spectra of CSPP NCs, while the XANES spectra of Pt-CNT are compared for reference. In a Pt L_3_-edge spectrum, the absorption edge (A) corresponds to the electron transition from 2p to 5d orbital, while the intensity (H_A_) is directly correlated to the unoccupied density of Pt 5d orbital (i.e., the amount of charge/electron relocation from surrounding atoms to Pt) and the extent of oxygen chemisorption (O^ads^) on the surface, therefore, can be correlated to the electrocatalytic activity of CSPP NCs [29]. Previously published studies frequently reported that the higher density of unoccupied Pt d-orbitals promotes the electrocatalytic activity of nanocatalysts [30]. As shown in Figure 3a, the CSPP-50 NC exhibits the highest white line intensity (H_A_), indicating the highest vacant Pt d-orbitals and, therefore, the highest ORR performance. Moreover, the similar inflection point position of CSPP NCs as that of Pt-CNT suggests a similar metallic characteristic in all the NCs. Furthermore, the extended X-ray absorption fine structure (EXAFS) analysis was employed to unveil the local coordination environment around Pt-atoms. Figure 3b depicts the *k*^3^ weighted Fourier-transformed (FT)-EXAFS spectra of CSPP NCs and Pt-CNT at Pt L_3_-edge, while the corresponding model simulated structural parameters are summarized in Table 1. Accordingly, CSPP-RT NC does not show a coordination number (CN) for the Co–Pt bond pair (i.e., CN_Co–Pt_ = 0), suggesting that due to limited galvanic replacement reaction, the Co atoms are mostly present in the core region instead of the surface. Meanwhile, the CNs for Pt–Pt (CN_Pt–Pt_), Pt–Pd (CN_Pt–Pd_), and Pt–Sn (CN_Pt–Sn_) bond pairs are 1.316, 0.795, and 2.274, indicating Pt-clusters (dimers; CN_Pt–Pt_-1.316) are grown on SnO_2_ encapsulated SnPd_2_ layer. Moreover, as compared to CSPP-RT NC, the CNs for Pt–Pt (CN_Pt–Pt_) and Pt–Pd (CN_Pt–Pd_) bond pairs are significantly increased, respectively, to 5.795 and 1.574 in CSPP-50 NC. These results together indicate the homoatomic clustering of Pt atoms (i.e., formation of atomic clusters) accommodated on defects of the SnPd_2_ alloy layer. The CN for Pt–Co bond pair is also observed in CSPP-50 NC, suggesting the exposure of Co-atoms to the surface and formation of Co–Pt alloys. Meanwhile, the decreased CN for the Pt–Sn bond pair (CN_Pt–Sn_ = 0.801) reveals that the SnO_2_ layer is disappeared in CSPP-50 NC and the small CN for Pt–Sn comes due to the Pt-clusters growth on the SnPd_2_ layer. All these observations confirm the formation of CoPt nanoalloys together with Pt clusters on the SnPd_2_ layer and CoO_2_ underneath. Further raising the temperature to 75 °C resulted in decreased CN for Pt–Pt, Pd–Pd, and Pt–Sn bond pairs, while increased CN for Pt–Co bond pairs. These scenarios ambiguously suggest that the formation of CoSnPdPt alloy structure capped with SnO_2_ layer on the surface (because the Pt–Sn CN is significantly decreased; therefore, the majority of Sn atoms formed SnO_2_ layer).

Figure 3c represents the XANES spectra of CSPP NCs at Co K-edge, which mainly shows two features. The pre-edge “X” around 7710 eV corresponds to the quadrupole-allowed 1s-to-3d electron transition, and the absorption edge “M” refers to the 1s-to-4p electron transitions [31,32]. As shown in Figure 3c, the standard CSPP NCs exhibit many intense pre-edges and mainly possess tetrahedral Co configuration, while the Co-CNT showing much-suppressed pre-edge possesses octahedral CO-configuration. Meanwhile, the inflection point position (denoted by I_f_) is shifted to higher energy values as compared to that of Co-foil, confirming that Co atoms are oxidized in all the samples. Given that the CSPP-50 NC exhibits the highest white line intensity (H_M_) among CSPP NCs, indicating the greatest extent of charge transfer from Co to neighboring atoms, which confirms that the inert SnO_2_ layer (barrier for charge transfer) is absent in CSPP-50 NC. Figure 3d shows the FT-EXAFS spectra of CSPP NCs at Co K-edge, where peaks “N” and “O” are contributions respectively from the Co–O and Co–Co bond pairs. The higher intensity of peak N confirms that Co is oxidized in CSPP NCs. Moreover, Figure 3e and Figure 3f, respectively, show the XANES and EXAFS spectra of CSPP NCs at Sn K-edge. The nearly unchanged inflection point position and absorption edge (in XANES spectra) along with intense peak O in EXAFS spectra (corresponding to the Sn–O bond) confirms that Sn is present in the form of SnO_2_ in all the samples.

The aforementioned structural configurations are further explored by using cyclic voltammetry (CV) analysis. Figure 4 compares the CV curves of CSPP NCs with commercial J.M.-Pt/C catalyst, measured in an N_2_-saturated 0.1 M KOH solution at a sweep rate of 20 mV/s. The CV curves of CSPP NCs as well as commercial J.M.-Pt/C catalyst exhibit three main regions including (i) hydrogen adsorption/desorption region (0.1 < E < 0.40 V vs. RHE) followed by (ii) double-layer region (0.4 < E < 0.50 V vs. RHE) and (iii) the oxygen adsorption (i.e., oxide formation in forward sweep) and subsequent reduction (reverse sweep) region above 0.50 V vs. RHE. As shown in Figure 4, the commercial J.M.-Pt/C catalyst exhibits two peaks in a forward sweep (E_H_^des-i^ and E_H_^des-ii^) as well as in a reverse sweep (E_H_^ads-i^ and E_H_^ads-ii^), which are, respectively, corresponding to the hydrogen desorption and adsorption on the open and close facets of Pt-crystal. On the other hand, the CSPP-RT and CSPP-75 NCs show smeared peak profiles below 0.40 V vs. RHE which can be attributed to the presence of an inert SnO_2_ layer. Notably, the CSPP-50 NC exhibits a relatively sharp E_H_^des-i^ peak as compared to CSPP-RT and CSPP-75 NCs, and as per published literature, such a peak corresponds to the strong H_2_ evolution activity by the atomic species [33,34]. Therefore, the presence of CoPt nanoalloys and Pt clusters are expected on the surface of CSPP-50 NC. Moreover, the position of the oxide reduction peak (i.e., E_O_^des^) is directly correlated to the energy barrier for ORR on the surface of the material. Accordingly, the oxide reduction peak (E_O_^des^) in a reverse sweep for CSPP-50 NC is shifted to the highest potential, suggesting the lowest energy for ORR as compared to CSPP-RT and CSPP-75 NCs.

Based on the above-discussed HRTEM, XRD, XAS, and CV results, the nanostructures of CSPP NCs have been proposed and shown in Figure 1. Accordingly, multilayered hierarchical structures with different configurations are formed in all the CSPP NCs. First, when the impregnation temperature is controlled to 25 °C, the Pt-clusters’ emerging SnO_2_ layer is formed over SnPd_2_ domains with CoO_2_ underneath (i.e., CSPP-RT). Further raising the impregnation temperature to 50 °C (i.e., CSPP-50 NC) resulted in a severe change in the configuration of surface as well as sub-surface regions. The cross-referencing results of HRTEM, XRD, and XAS consistently confirm that due to enhanced kinetics of galvanic replacement reaction, Co atoms are exposed to surface as well as sub-surface regions, and the inert SnO_2_ layer vanished. Moreover, combining the results of CV analysis with microscopy and spectroscopic observation, the formation of CoPt nanoalloys and Pt-clusters on the surface of CSPP-50 NC is confirmed. Moreover, the results of physical structure inspections suggest the formation of SnO_2_-capped multi-metallic CoSnPdPt alloy over CoO_2_ support in CSPP-70 NC.

The ORR activities of CSPP NCs were evaluated by linear sweep voltammetry (LSV) in an O_2_-saturated 0.1 M KOH electrolyte. The LSV curves of CSPP NCs and commercial J.M.-Pt/C catalyst are represented in Figure 5a, while the corresponding electrochemical parameters are summarized in Appendix A. Accordingly, the CSPP-50 NC achieved the highest half-wave (E_1/2_) (0.894 V) and onset (E_OC_) (0.946 V) potentials among experimental NCs (Figure 5b). Previously published studies frequently reported that the E_1/2_ and E_OC_ are key performance matrices for ORR and are in close resemblance with the energy barrier for initiating the ORR [35]. The high E_1/2_ and E_OC_ refer to the lower energy barrier for initiating the ORR on the material’s surface. Therefore, the highest E_1/2_ and E_OC_ of the CSPP-50 NC suggest the lowest energy barrier for oxygen reduction from its surface and are consistent with E_O_^des^. Peak position in the CV curve. Furthermore, the LSV polarization curves were used to calculate the kinetic current densities (*J_K_*) of experimental samples (Figure 5c) and further normalized at 0.85 V vs. RHE with respect to Pt and Pt + Pd loadings to obtain the mass activities (MA)s (Figure 5d). Unsurprisingly, the CSPP-50 demonstrated the highest MA of 4330 mA mg_Pt_^−1^ at 0.85 V vs. RHE, which is 78-folds improved as compared to commercial J.M.-Pt/C catalyst. In clarifying the effect of Pt-decoration on the CSP surface, the ORR performance of CSPP NC is compared with CSP NC and shown in Appendix A, where the CSPP NCs exhibit far better ORR performance as compared to CSP NC.

## 4. Conclusions

The present study implements a versatile and effective approach for controlling the configuration of quaternary metallic nanocatalysts (NC)s via marginally changing the impregnation temperature. By controlling the impregnation temperature to 50 °C, the Pt-clusters and CoPt nanoalloys anchored CoO_2_@SnPd_2_ hierarchical structure with multiple metal-to-metal oxide interfaces are prepared (namely CSPP-50). As-prepared CSPP-50 NC demonstrated the mass activity of 4330 mAmg_Pt_^−1^ at 0.85 V vs. RHE in alkaline ORR, which is 78-folds increased as compared to J.M.-Pt/C catalyst with 20 wt.% Pt (mAmg_Pt_^−1^). On the other hand, when the impregnation temperature is controlled to room temperature, the Pt-clusters emerged SnO_x_ layer is formed over CoO_2_@SnPd_2_, while the Sn-oxide layer covered CoO_2_@CoSnPdPt alloy structures are formed for 75 °C impregnation temperature. The experimental observations indicate that the potential synergism between surface decorated Pt-clusters, CoPt-nanoalloys, and adjacent SnPd_2_ domain enables the high ORR performance of CSPP-50 NC, where Pt-clusters deliver ideal adsorption energy for O_2_ splitting (i.e., O–O bond breaking) and CoPt-nanoalloys along with SnPd_2_ domain facilitates the subsequent desorption of hydroxide ions (OH^−^).

## Data Availability

The data presented in this study are available on request from the corresponding author.

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
