# Peer review of "Co-Existence of Atomic Pt and CoPt Nanoclusters on Co/SnO_x_ Mix-Oxide Demonstrates an Ultra-High-Performance Oxygen Reduction Reaction Activity"

_nanomaterials, 2022, doi:10.3390/nano12162824_

Round 1
Reviewer 1 Report
The article "Co-existence of atomic Pt and CoPt nanoclusters on Co/SnOx mix-oxide Demonstrates an Ultra-High-Performance Oxygen Reduction Reaction Activity" describes the synthesis of new composite catalysts with a low platinum content, the composition and structure of the obtained materials, as well as their activity in ORR. The strong part of the article is undoubtedly a detailed and comprehensive study of the obtained materials structure. However, it is important to note that the representation of the proposed structure evolutions for CSPP NCs is a simplified model and the structure of the resulting material is extremely complex. The topic of research is undoubtedly relevant, since the production of efficient platinum-containing catalysts with a low platinum content is necessary for the development of hydrogen energy. Nevertheless, it is worth noting a number of significant comments on the article:
Nevertheless, it is worth noting a number of significant comments on the article:
1. The paper does not present the actual composition of the obtained materials.
2. Rationing the activity of catalysts containing Pt and Pd, only for platinum is impractical. It is necessary to clearly indicate the content of Pt and Pd in each material directly in the article. Reference in abstract “Such a nanocatalyst (NC) outperforms the commercial Johnson Matthey-Pt/C (J.M.-Pt/C; 20 wt.% Pt) catalyst by 78-folds with an outstanding mass activity (MA) of 5,245 mAmgPt-1 at 0.85 V vs RHE in an alkaline medium (0.1 M KOH)" is incorrect because it does not take into account the contribution of palladium to the catalytic activity.
3. The section “Preparation of the Electrode and the Calculation for ORR Mass Activity” from “non-published” should be moved to the text of the article in the section “Experimental Procedures”.
4. The article presents the results of TEM, however, for the studied materials, the average size of nanoparticles was not determined and the histogram of the particles size distribution was not presented.
5. To assess the activity of a catalyst, it is important to determine the value of ESA and normalize the activity to the value of ESA.
6. Why are the bonds of metals with light atoms not presented in table 1, for example, Pt-O, Co-O, etc.?
Reviewer 2 Report
It is good and exciting work in the field of Oxygen Reduction Reaction. However, authors are requested to improve it further by considering the following comments.
1. XRD should marked for convenience of the readers.
2. Authors should provide some post analysis results of the catalyst.
3. Authors are encouraged to do any ex-situ characterization and provide some insights for the charge storage mechanism.
4. It is important to cite some relevant and recent literature, such as.: Chemical Engineering Journal, 440, 135928. Energy Storage Materials, 45, 301-322. ACS applied materials & interfaces, 13(19), 23191-23200.
5. English has to be improved.
Round 2
Reviewer 1 Report
The authors have made the necessary changes and the article can be accepted for publication.
Reviewer 2 Report
Can Accept in its current form.